# Long-term coal gangue dump regulates bacterial communities in different disturbance areas: Evidence mostly from diversity and network complexity

**Bianhua Zhang**[1,2], **Dongsheng Jin**[2,3,4,5]*, **Qiang Zhang**[2,3,4,5]*, **Huijuan Bo**[2,3,4,5], **Wei Wang**[2,3,4,5]

1 Xinzhou Normal University, Xinzhou, P. R. China, 2 Key Laboratory for Soil Environment and Nutrient Resources in Shanxi Province, 3 Shanxi Agricultural University, Taiyuan, P. R. China, 4 Key Laboratory for Farmland Fertility Improvement of Eastern Loess Plateau (Jointly-founded by MARA and Shanxi Province), Ministry of Agriculture and Rural Affairs, P. R. China, 5 Engineering Technology Innovation Center for Ecological Protection and Restoration in the Middle Yellow River

* sxdxjds@126.com (DJ); sxsnkytfs@163.com (QZ)

## Abstract

In order to clarify the effects of long-term coal gangue(CG) dump on the surrounding soil bacterial community structure, we selected the CG dump formed during the mining of Tunlan coal mine in Gujiao city, Shanxi province in China as the study area to conduct a comprehensive study, the experimental design included six distinct zones: control soil area with no impaction (NC), undisturbed control sediment area (NL), atmospheric dry and wet deposition area (MC), upstream (MLS), midstream (MLZ) and downstream (MLX) in the leachate flow area (LFA), Using high-throughput sequencing technology and related software analysis, we obtained the following key findings: The heavy metal contents of Cr and Cd were different significantly in MC and NC ($p < 0.05$),Cr (90.18 mg·kg-1) in MC was higher than that in NC (65.29 mg·kg-1) ($p < 0.05$), while Cd (0.09 mg·kg-1) was lower than that in NC (0.14 mg·kg-1) ($p < 0.05$), and there was no significant differences in Cu, Zn, As and Pb between MC and NC ($p > 0.05$). All the heavy metal contents in MLS were highest significantly except Cd among NL and LFA. Shannon and Chao1 indices in NC were significantly higher than those in MC ($p < 0.05$), In LFA, Shannon and Chao1 indices of MLX were the highest, while MLS was significantly lower than NL ($p < 0.05$). The relative abundance of bacteria more than 40% in MC and NC was Actinomycetes (42.06%−42.38%), and while was Proteobacteria (40.66%−50.77%) in NL and LFA. Bacterial communities in different disturbed areas were significantly correlated with As, Pb, Cu, Cd, TP, SOC and EC in the soil, among which SOC contributed most about 40.1%. Molecular ecological network showed that the interactions among bacterial taxa in MC and LFA were mainly in a positive synergistic development, the bacteria with higher relative abundance may not be the key node in the bacterial

**Data availability statement:** All relevant data are within the paper and its Supporting information files.

**Funding:** This study was funded by the National Key R&D program of China, 2020YFC1806501-2, Dongsheng Jin; Laboratory platform foundation project, 2025-HSWY-XB-106, Dongsheng Jin; Key R&D program project of Xinzhou municipal science and technology bureau, 20240301, Dongsheng Jin; Major project of science and technology in Shanxi province, 202201140601028, Qiang Zhang; Supported by fundamental research program of Shanxi province, 202203021221223, Bianhua Zhang; Open fund project from key laboratory for soil environment and nutrient resources in Shanxi province, 2024003, Bianhua Zhang.

**Competing interests:** The authors have declared that no competing interests exist.

molecular ecological network, while bacteria with the lower relative abundance might have been. The bacterial community structure of MC was more complex than NC because of fewer nodes and modules but more connections. The positive connection proportion and modules of bacteria in LFA was higher than that in NL, while the aggregation coefficient decreased. The average path distance (5.09) in MLS was the shortest, indicating the bacteria in MLS were most environmental sensitive to the external environment with rapid community response to disturbances. Our results revealed the changes in the bacterial community and the main environmental driving factors under disturbance of CG dump, this information provides a theoretical basis for ecological environment management.

## 1. Introduction

The main components of coal gangue (CG)include $SiO_2$, $Al_2O_3$, $Fe_2O_3$, MgO, CaO, $SO_2$, MnO, coal, and toxic heavy metal elements such as Pb, Cd, Hg, As, and Cr [1], and its surface bacterial diversity is relatively low [2]. It primarily originates from processes such as coal mining, excavation, washing, and open-pit stripping, and it accounts for 10%−15% of coal production [3]. Currently, the stockpile of CG in China has reached 5 billion tons [4], which is typically piled around mining areas, forming large CG mountains [5,6]. Research has indicated that modified CG can be used as a plant substrate to enhance soil biological activity [7], improve soil pore structure, and increase the content of organic carbon and its components in soil [8]. Adding CG as fertilizer to the soil can boost the number of soil bacteria and fungi [9], increase the soil's available phosphorus (AP)content [10], and reduce the migration rate of heavy metals in the soil [11].

Long-term open-air stacking of CG will produce many harmful soluble substances [12], resulting in an increase in heavy metal content in the soil [13,14], and a decrease in surface water quality [15–17]. Some studies have shown that the content of heavy metals in soil decreases with the increase of distance from CG storage area, or increases first and then decreases [18,19]. Generally, the surface layer is higher than the bottom layer [13,20]. And a similar phenomenon occurs in the surrounding surface water, where the closer to the gangue pile is, the worse the water quality [21]. It shows that the mechanism of the impact of CG stockpiles on the environment includes atmospheric dust reduction and leaching process [22].

Microbial communities are sensitive indicators of soil quality [23], and are drivers of ecosystem functioning, involved in carbon and nitrogen cycling, soil pollutants, and soil structure stability [24]. Studies have shown that the main phyla of bacteria in weathered CG are Proteobacteria, Actinobacteria, and Chloroflexi [25], and the bacterial taxa are significantly affected by EC and particle size [26]. Some studies have shown that soil microbial diversity varies significantly with distance from the CG stockpile [27]. Microbial biomass decreases with increasing distance from the CG stockpile [28]. In general, most of these studies focus on the microbial response under a single disturbance. There is a lack of systematic research on the response

mechanism of soil microbial communities in different disturbance areas, such as the differences in the composition and diversity of soil bacterial communities in atmospheric dust areas and leachate flow areas? Which environmental factors play a decisive role?

In this study, it was hypothesised that the heavy metals and nutrient contents in the soil in the disturbed area of the CG dump would be different, and they might interact with each other, which would affect the structure and diversity of soil bacterial communities. In order to test this hypothesis, this paper selected the CG dump in Tunlan mining area of Gujiao City, Shanxi Province as the study area, collected soil samples from different disturbance areas around the dump, and used high-throughput sequencing technology combined with molecular ecological network to study the response mechanism of soil microorganisms around long-term CG dump. The main research contents include: 1) using 16Sr DNA technology to study the diversity, structure, composition and interrelationship Interactions among bacterial taxa in the disturbed areas of CG dump; 2) Analyze the differences in heavy metal content and physicochemical properties in each disturbance zone; 3) to explore the correlation between bacterial community structure and soil environment.

The rest of this study is structured as follows: Part 2 describes study areas, soil sampling, sample determination and data statistical analysis; Part 3 and Part 4 present the research results and discuss them using related references, including analysis of soil properties, bacterial community structure and diversity, correlation between them around CG dump; Part 5 is the conclusion and limitations of the study, providing context for the findings and suggesting directions for future research.

The contributions of this study are: (1) Provides critical empirical evidence on how long-term coal gangue (CG) storage alters soil ecosystems, specifically quantifying heavy metal and its heterogeneous effects on bacterial communities across distinct disturbance areas for ecological risks in mining-impacted regions; (2) By molecular ecological network analysis challenges conventional diversity-centric assessments and highlights the functional importance of bacterial taxa in ecosystem resilience; (3) By finding out the primary driver from soil environmental factor effected on microbial community structure as offers actionable information for bioremediation efforts.

## 2. Materials and methods

### 2.1 Study area

The study was conducted in Tunlan mining area of Gujiao city, Shanxi Province, China, located at latitude 37°40.1'N-38°10.5'N and longitude 111°43.5'E-112°21.3'E. It is the temperate continental climate, the annual sunshine number is 2808 hours, the annual average precipitation and evaporation are 420 mm and 1025 mm respectively. It is windy in winter and spring, with a maximum wind speed of 7.2 m/s, mostly in the north-west direction.

The CG dump form is an open coal gangue storage. The total storage is about 32 hm². Its main chemical composition is: $Al_2O_3$, $SiO_2$ and $Fe_2O_3$, also including a variety of metal elements such as Zn, As, Hg, Mn, Fe, Cr and so on. According to the landform and climate conditions around the CG dump in Tunlan mining area, we found that the disturbance of the CG dump to the surrounding soil is mainly filtered by atmospheric dustfall and rainfall eluviation. Therefore, the sampling point is set within 300 m range of the atmospheric dry and wet deposition area under the prevailing wind of the CG dump, we marked it as MC; the soil in the area without CG interference was marked as NC; the upper, middle and lower reaches was labeled as MLS, MLZ and MLX respectively in the leach flow area (LFA), and the undisturbed sediment area was marked as NL(Fig 1).

### 2.2 Soil sampling

In July 2022, using five-point sampling method, we collected the surface soil within 0–20 cm in each area, and removed gravel, dead leaves and other impurities in the collected samples. Part of the samples were put into sterilized sealed bags for microbial DNA sequencing analysis, and some were brought back to the laboratory to dry and sieved for analysis of physical and chemical properties and heavy metal content.

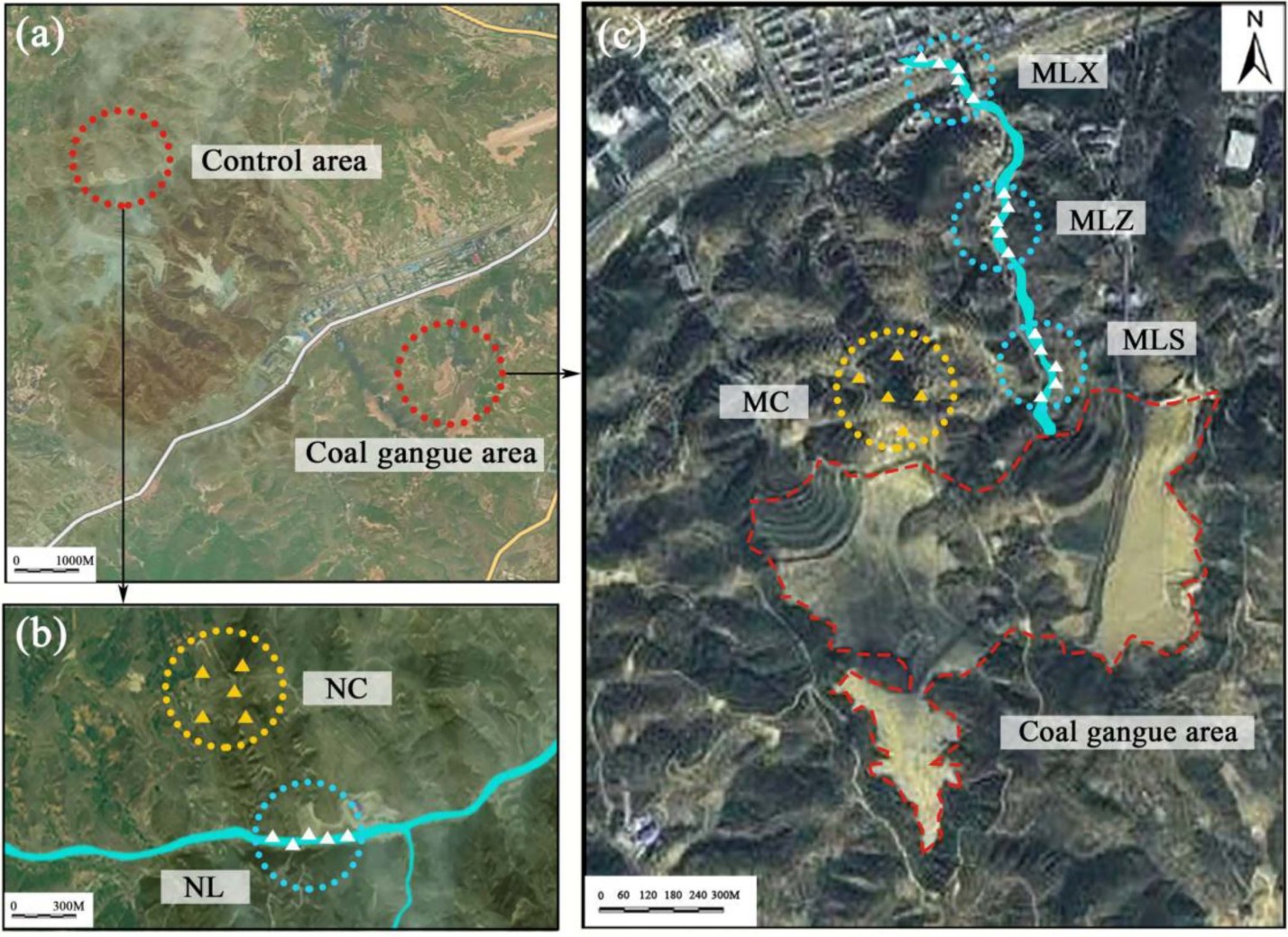

**Fig 1. Location and division of the study area.** Note: Circles are the sampling areas of each treatment, triangles are the distribution of each sampling. NC refers to control soil area with no impaction, NL refers to undisturbed control sediment area, MC refers to atmospheric dry and wet deposition area, MLS, MLZ and MLX refers to upstream, midstream and downstream in the leachate flow area (LFA) respectively. The same below.

## 2.3 Sample determination

Soil pH value was determined by pH meter (PHS-3C) with water to soil ratio of 2.5:1 [29]. The electrical conductivity (EC) was determined by the method of 5:1 water to soil ratio with FE30 Plus instrument [30]. Soil available potassium (AK) was determined by CH3COONH4 extraction with flame photometer (FP6400A) [31]. The available phosphorus (AP) was determined by UV spectrophotometer (UV-1900SPC) with $NaHCO_3$ molybdenum extraction [32]. Total potassium (TK) was determined by flame photometer (FP6400A) using NaOH melting method [33]. Soil organic carbon (SOC) was determined by potassium dichromate volumetric method [34]. Total phosphorus (TP) was used by NaOH molybdenum blue colorimetry and ultraviolet spectrophotometer (UV-1900SPC) [35]. Total nitrogen (TN) was determined using a nitrogen analyzer (Sano KT8400) using the Kjeldahl nitrogen determination method [36]. Soil heavy metal content including Cr, Cu, Zn, As, Cd, and Pb was determined using an inductively coupled plasma emission spectrometer (ICP-OES, Avio200, PerkinElmer, Inc., USA) [37].

16Sr DNA sequencing: 0.5 g soil samples were weighed and the soil microbial DNA was extracted and purified using the OMEGA soil DNA kit (D5625-01) (OMEGA Bio-Tek, Norcross, GA, USA). The $2 \times 250$ bp double-ended sequencing was performed using the NovaSeq6000 SP kit (500 cycles) on the Illumina NovaSeq machine. PCR amplification of V3-V4 highly variable region of 16SrRNA was performed using 338F (5' -barcode +ACCTACGGGAGGCAGCA-3') and 806R (5' -GACTACHVGGGTW TCTAAT-3 ') priors.

The PCR process is as follows: (1) pre-denaturation at 98°C for 30 s and amplification cycle begins. (2) The template is denatured at 98°C for 15 s, then the temperature is lowered to 50°C for 30 s to allow the primer and template to be fully annealed. (3) Hold at 72°C for 30 s until primer extensions are formed on the template, DNA is synthesized, and a cycle is completed. The cycle is repeated 25–27 times to accumulate a large number of amplified DNA fragments. (4) The product is kept at 72°C for 5 minutes to fully extend and stored at 4°C. Quantification of PCR products (PicoGreen dsDNA) was performed on a microplate reader (BioTek, FLx800) using Quant-iT, and libraries were constructed using Illumina TruSeq Nano DNA LT library preparation kit. The sequencing was carried out on the Illumina MiSeq platform of Shanghai Peisenol Microbiological Technology Company. QIIME2 (2019.4) software was used to trim sequence fragments, denoise, splice and remove chimeras, and generate an operational taxon (ASV) with sequence similarity threshold of 100%.

## 2.4 Data statistical analysis

A one-way ANOVA was conducted to determine the effects of CG dump on soil physicochemical properties and Alpha diversity index (Chao1, Shannon) using SPSS Statistics (Version 26.0, SPSS Inc., Chicago, IL, USA), and means were declared significant at ($p < 0.05$) under Tukey's Honesty Test [38]. Principal Coordinate Analysis (PCoA) based on the Bray-Curits distance algorithm was used to assess similarities in bacterial community structure among samples. Canoco5.0 (Canoco5 Support Site www.canoco5.com) software [39] was used for redundancy analysis to obtain the correlation between bacterial community structure and soil environmental factors. Based on random matrix theory (RMT), phylogenetic molecular ecological networks and their topological nature was constructed using the online Molecular Ecological Network Analysis (MENA) pipeline (http://ieg4.rccc.ou.edu/mena/) [40], and based its built-in Spearman correlation matrix, and the network results were visualized using Cytoscape (Version 3.9.1, https://cytoscape.org/) and its built-in plugin (Network Analyzer) [41]. Origin (Version9.0, Origin Lab Corporation, Canada) is used to visualized topological roles, which are characterized using inter-module connectivity (Pi) and intra-module connectivity (Zi), nodes are divided into four categories: Network Hub (Pi > 0.62 and Zi > 2.5), Module Hub (Pi ≤ 0.62 and Zi > 2.5), Connector Node (Pi > 0.62 and Zi ≤ 2.5), and Peripheral Node (Pi ≤ 0.62 and Zi ≤ 2.5) [42].Generally Zi ≥ 2.5 or Pi ≥ 0.62 node as the key node, in their respective modules and modules within and between the connection with other nodes play an important role.

## 3. Results and analysis

### 3.1 Analysis of soil properties around CG dump

As can be seen from Table 1, the contents of Cr and Cd in the soil of MC were significantly different from that of NC. The content of Cr in MC was significantly increased by 24.89 mg•kg⁻¹ compared with that of NC, while the content of Cd was significantly decreased by 0.05 mg•kg⁻¹ compared with that in NC ($p < 0.05$). There was no significant difference in Cu, As, Pb and Zn between MC and NC. Cr, Cu, As, Pb and Zn in MLS were significantly higher than NL, MLZ, MLX, and MC. The content of As in MLZ and MLX was significantly higher than that in NL. It explained the heavy metal contents had big differences in the different disturbed area.

It can be seen from Table 2 that there were great differences in physical and chemical properties of soil in different disturbed areas. The contents of EC, SOC and TN in MLS were significantly higher than those in other disturbed areas, while TK and pH values were lowest. There were no significant differences in SOC and TK contents between MC and NC ($p > 0.05$), while TN, TP and AK contents in MC were significantly lower than those in NC ($p < 0.05$), and AP contents were

**Table 1. Content of heavy metals in soil in different disturbed areas of CG dump (mg· kg $^{-1}$).**

| Treatment | Cr | Cu | As | Cd | Pb | Zn |
|---|---|---|---|---|---|---|
| NC | 65.29±5.42c | 24.36±0.82b | 11.94±0.32ab | 0.14±0.02a | 19.66±0.51c | 67.25±1.9c |
| MC | 90.18±22.17b | 26.19±2.74ab | 12.67±1.14ab | 0.09±0.04bc | 19.02±1.3c | 70.85±6.16c |
| NL | 78.01±12.44bc | 24.42±0.78b | 7.48±0.17d | 0.1±0.03abc | 21.72±0.47c | 69.17±1.8c |
| MLS | 110.75±11.51a | 28.67±1.29a | 14.22±0.51a | 0.07±0.06c | 35.36±1.32a | 97.66±4.22a |
| MLZ | 81.75±9.84bc | 23.73±0.53b | 8.97±0.23c | 0.13±0.02ab | 25.52±1.25b | 81.43±2.69b |
| MLX | 75.58±10.03bc | 25.17±1.46b | 11±0.68bc | 0.13±0.01ab | 26.92±0.79b | 72.88±2.96bc |

Note: The data in the table are mean±standard error, different lowercase letters within a column indicate significant differences at the 5% level. Significant difference was observed in different disturbed areas of soils.

**Table 2. Physical and chemical properties in soil in different disturbed areas of CG dump.**

| Treatment | pH | EC (ms·cm$^{-1}$) | SOC (g· kg $^{-1}$) | TN (%) | TP (g· kg $^{-1}$) | TK (g· kg $^{-1}$) | AK (mg· kg $^{-1}$) | AP (mg· kg $^{-1}$) |
|---|---|---|---|---|---|---|---|---|
| NC | 8.60±0.06a | 91.34±0.95f | 7.77±5.22e | 0.08±0.00c | 0.64±0.02a | 16.16±0.40a | 75.51±16.72a | 4.00±0.63b |
| MC | 8.11±0.07b | 627.46±37.15d | 13.19±2.35de | 0.05±0.01e | 0.24±0.20c | 14.20±3.48ab | 36.63±0.8b | 5.20±0.69a |
| NL | 8.17±0.05b | 285.60±10.70e | 14.78±3.49 cd | 0.06±0.01d | 0.41±0.06b | 13.39±0.87bc | 52.51±32.33ab | 3.81±0.74b |
| MLS | 6.43±0.20d | 2324.60±87.05a | 27.44±4.45a | 0.12±0.00a | 0.35±0.04bc | 10.39±1.60d | 51.56±9.3ab | 2.01±0.35c |
| MLZ | 7.29±0.10c | 937.4±129.77c | 20.00±3.32bc | 0.09±0.00b | 0.26±0.08c | 11.02±1.75 cd | 37.27±1.89b | 1.77±0.39c |
| MLX | 7.26±0.49c | 1320.6±37.68b | 24.17±2.35ab | 0.08±0.00bc | 0.46±0.01b | 14.09±0.33ab | 48.36±4.9b | 2.54±1.03c |

Note: The data in the table are mean±standard error, different lowercase letters within a column indicate significant differences at the 5% level. Significant difference was observed in different disturbed areas of soils.

significantly higher than those in NC (p < 0.05). In the leaching flow area, SOC and TN contents in MLS were significantly higher than those in NL and MLZ, while TK and AP contents were significantly lower than those in NL (p < 0.05), and AK content in LFA were not significantly different (p > 0.05). The value of pH in MC was significantly lower than that in NC, and its value in MLZ and MLX was significantly lower than that in NL (p < 0.05), EC value in MC, MLS, MLZ and MLX was significantly higher than that in NC and NL (p < 0.05). It indicated that long-term accumulation of CG could decrease pH and increase the EC in the soil.

As can be seen from Fig 2, the main bacterial phyla in the soil in the different disturbed areas are Proteobacteria (22.77%−50.77%), Actinobacteria (12.05%− 42.38%), Chloroflexi (5.62%−12.71%), Acidobacteria (5.44%−12.76%), Bacteroidetes (0.89%−8.88%) etc. Relative abundance of Proteobacteria and Actinobacteria was more than 20% in MC and NC. While Proteobacteria with relative abundance greater than 20% could be only found in the soil of NL, MLS, MLZ and MLX, and their relative abundance was significantly higher than that of MC and NC. The results indicated that the soil bacterial community structure changed significantly after the leaching of CG. We found the relative abundance of Actinobacteria and Bacteroidetes in MLS decreased significantly while the abundance of Chlorobacteria and Acidobacteria increased significantly compared with that in NL.

As can be seen from Fig 3, the proportion of the top 20 bacterial genera with higher relative abundance was the smallest in NL and the highest in NC, which in MC was higher than the leach ditch, and that in MLS was higher than MLZ and MLX. The relative abundance of KD4–96 in MLS was the highest, about 5%, and that of Subgroup_6 was the highest in NC. CG storage decreased its relative abundance after atmospheric dustfall and leaching, which was only about 2% in MLZ. The relative abundance of Sphingomonas in MC was higher than that in NC and LFA. In general, we found there were significant differences in soil bacteria genera in different disturbed areas.

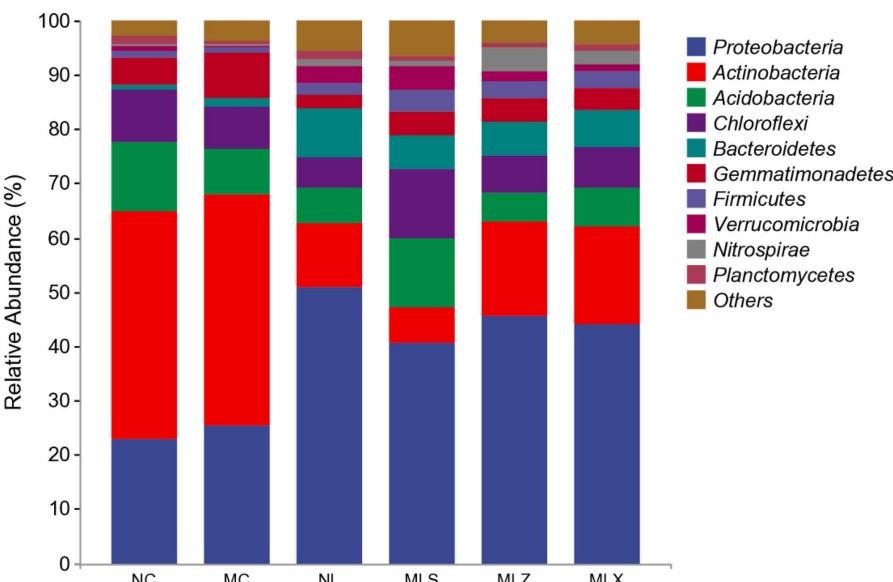

**Fig 2. Relative abundance of soil bacterial phyla in the disturbed areas of CG dump (%).**

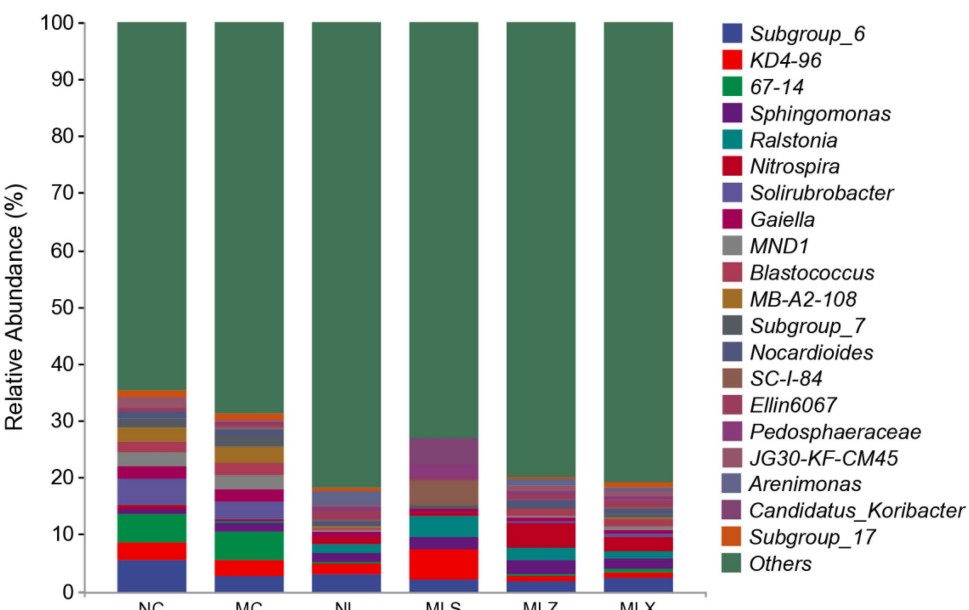

**Fig 3. Relative abundance of soil bacterial genus in the disturbed areas of CG dump (%).**

### 3.2 Analysis of soil bacterial Alpha and Beta diversity around CG dump

Chao1 index characterizes species richness, and Shannon index characterizes species diversity and evenness. In the study, we used Chao1 and Shannon index to characterize bacterial Alpha diversity. As can be seen from Fig 4, the order of Chao1 index and Shannon index among different disturbed areas is MLX > NC > NL > MLZ > MC > MLS, and there were significant differences in soil bacterial community diversity among different disturbed areas. Chao1 and Shannon index in

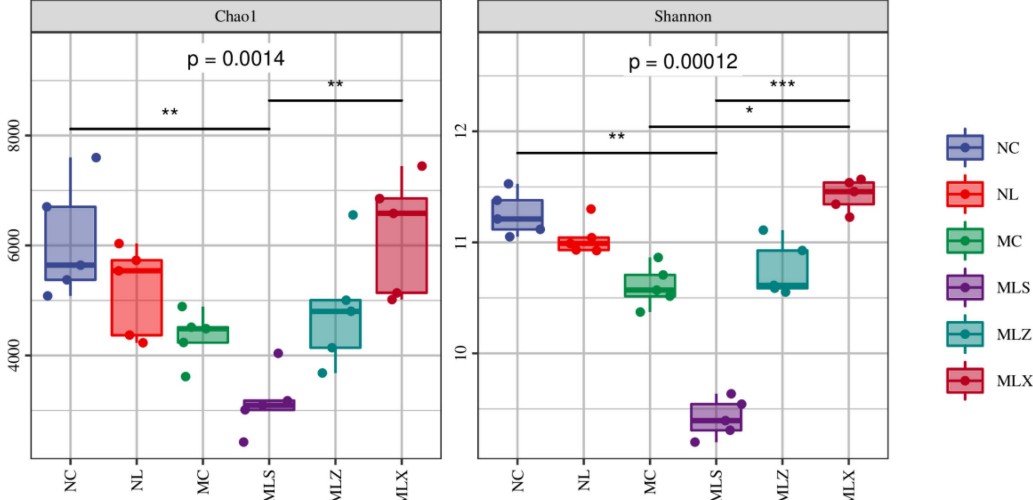

**Fig 4. Chao1 and Shannon indexes of soil bacteria in different disturbed areas of CG dump.** Note: The P-value represents the overall intergroup differences from the Kruskal- Wallis test, the significance level markers for pairwise differences obtained using Dunn's post hoc test (***represents p < 0.001, **represents p < 0.01, * represents p < 0.05).

the soil in MC were significantly lower than those in NC (p < 0.05), and the difference between NL and LFA was also obvious (p < 0.05), bacteria diversity in MLX was more abundant than that in MLS.

From Fig 5, we could find the soil in NC is clustered with MC in the same category, and LFA is clustered with NL in the same category. Moreover, there is a significant difference between MC and NC, and there is little difference between MLZ and MLX, while there is a big difference between NL and MLS. In order to study the differences between MC and NC, as well as NL and LFA, PCoA method was used for analysis. It can be found from Fig 6 that MC and NC are separated by the PCo1 axis, indicating a large difference between them. MLZ and MLX in the second quadrant, MLS in the first quadrant, and NL in the third quadrant, indicating that there are significant differences among NL, MLS, MLZ and MLX.

### 3.3 Correlation analysis between soil bacterial community and physicochemical properties

As shown in Fig 7 and Table 3, the total explanatory capacity of soil physical and chemical properties to bacterial community was 82.97%, and the explanatory capacity of axis I and axis II were 62.14% and 19.63% respectively. Among them, the contribution rate of SOC was the largest, accounting for 40.1%, and the contribution rate of As and Pb in heavy metals was relatively large, are 18.8% and 11.5% respectively. It could also be seen from Fig 7 that As, Cd and TP had a great influence on the soil bacterial community in MC, while Cu, Pb, SOC and EC had a great influence on MLS, and Cd influenced much on MLZ and MLX.

### 3.4 Molecular ecological network structure of soil bacteria around CG dump

The co-occurrence pattern of soil bacteria community was greatly influenced by the atmospheric dust fall and leachate infiltration of CG. Molecular ecological network was constructed based on the sequencing results of bacteria in different disturbed areas, and the threshold was automatically determined to be 0.97 or 0.98, with $R^2$ values greater than 0.80. As shown in Table 4 and Fig 8, the positive correlation between bacteria in different disturbed areas accounted for more than 58%, indicating that cooperative cooperation was dominant and competition was weak. The network nodes and the number of connections in MC and NC were smaller than those in MLS, MLZ, MLX and NL. Compared with NC, the number of nodes was smaller in MC, only 317, but more connections and higher positive correlation ratio, indicating that

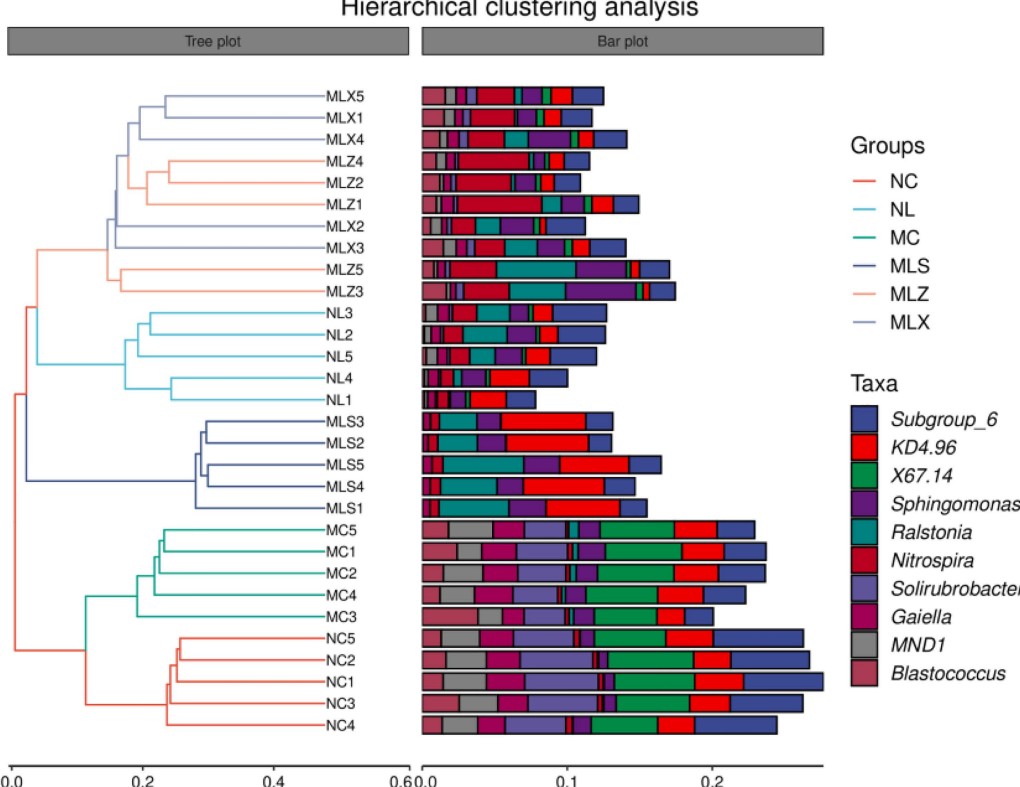

**Fig 5. Beta diversity of soil bacteria in different disturbed areas of CG by hierarchical cluster analysis.**

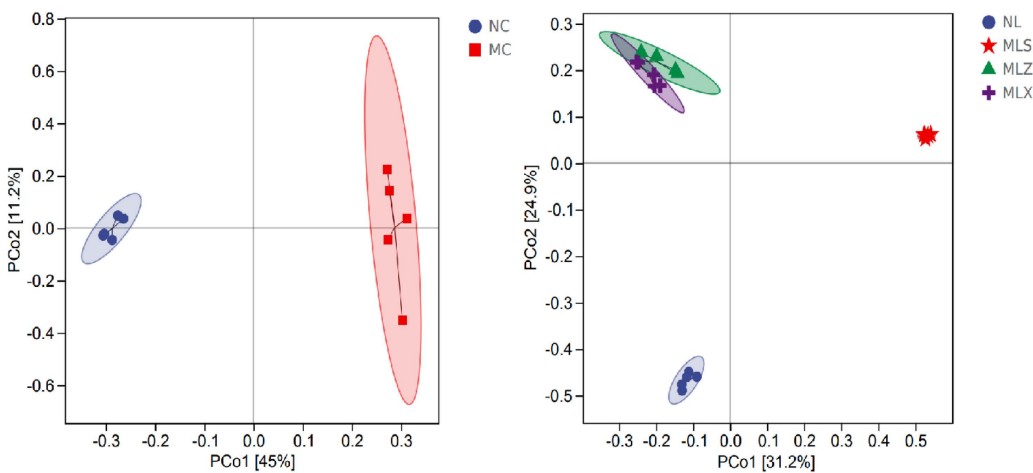

**Fig 6. Beta diversity of soil bacteria in different disturbed areas of CG by PCoA.**

the synergistic coexistence of bacteria in MC was stronger than NC, and the network was more complex. The positive connection proportion of bacteria and the number of modules in MLS, MLZ and MLX was higher than that of NL while the clustering coefficient decreased. The average path distance in MLS was the smallest in LFA.

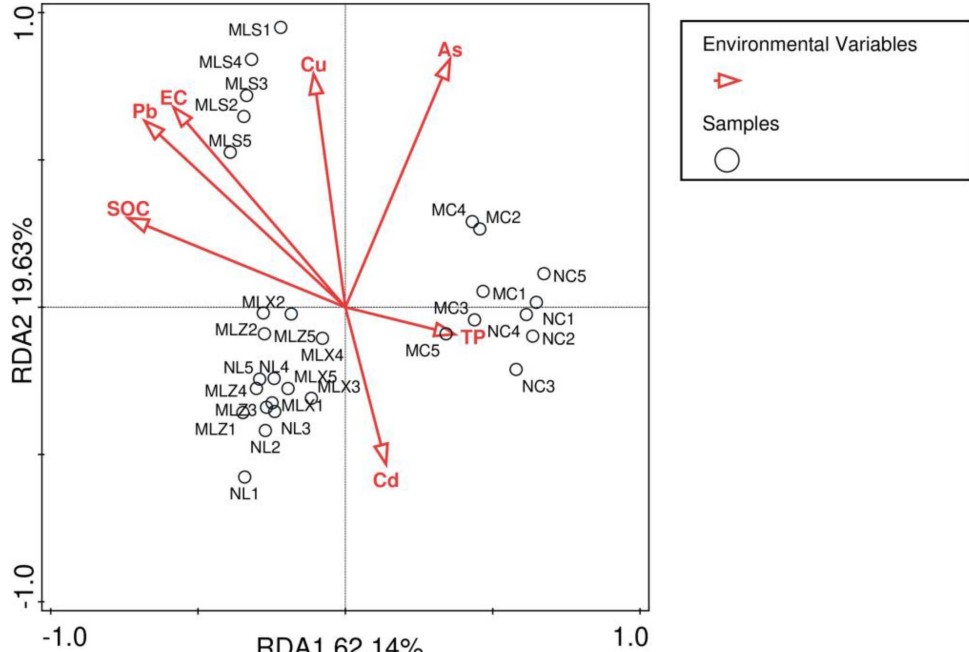

**Fig 7. Relationship between bacteria and physical and chemical properties in the soil by redundancy analysis.**

**Table 3. Results of redundancy analysis.**

| Name | Explains % | Contribution % | pseudo-F | P |
|------|-----------|----------------|----------|---|
| SOC | 35.8 | 40.1 | 15.6 | 0.002 |
| As | 16.8 | 18.8 | 20.4 | 0.002 |
| Pb | 10.3 | 11.5 | 5.1 | 0.012 |
| EC | 9.3 | 10.5 | 5.4 | 0.008 |
| TP | 8 | 8.9 | 5.4 | 0.006 |
| Cu | 3.3 | 3.7 | 4.7 | 0.004 |
| Cd | 2.2 | 2.4 | 3.4 | 0.022 |

As can be seen from Fig 9, there was no network hubs, most nodes in different disturbed areas are peripheral nodes, connector was only distributed in NL and MLZ, and there were 3, 5, 17, 12, 12, and 13 module hubs in NC, MC, NL, MLS, MLZ and MLX respectively. As shown in Table 5, these key nodes mainly concluding Proteobacteria Cyanobacteria, Chlamydiae, Dependentiae played an important connecting role in the bacterial network. We also found that in NC and MLS, the relative abundance of Actinobacteria was 42.06% and 6.42%, but there is no module hub that plays an important role under this phylum. However, Chlamydiae was the key module hub although the relative abundance was only 0.19% in MLS.

## 4. Discussion

### 4.1 Influence of CG dump on surrounding soil properties

Atmospheric particulate matter including heavy metals tends to deposit on the ground surface along the prevailing wind direction, leading to soil contamination [43]. Long-term open-pit CG dump has been identified as a primary source of heavy metal pollution in surrounding soils in the mining area [44]. Shang et al. (2023) reported that the surrounding soil

**Table 4. Parameters of soil bacterial molecular ecological network in different disturbed areas of CG dump.**

| Network indexes | NC | MC | NL | MLS | MLZ | MLX |
|---|---|---|---|---|---|---|
| Total nodes | 333 | 317 | 700 | 542 | 707 | 580 |
| Total links | 354 | 534 | 2429 | 733 | 1949 | 922 |
| Module number | 67 | 45 | 62 | 99 | 102 | 113 |
| Positive | 58.19% | 65.86% | 58.22% | 65.07% | 62.43% | 72.80% |
| Negative | 41.80% | 36.14% | 41.78% | 34.93% | 37.57% | 27.13% |
| R square of power-law | 0.88 | 0.88 | 0.85 | 0.82 | 0.85 | 0.86 |
| Average degree | 2.13 | 3.37 | 6.94 | 2.85 | 5.51 | 3.18 |
| Average path distance | 4.58 | 5.53 | 7.73 | 5.09 | 7.70 | 6.76 |
| Average clustering coefficient | 0.25 | 0.23 | 0.44 | 0.29 | 0.36 | 0.25 |
| Modularity | 0.95 | 0.81 | 0.86 | 0.93 | 0.88 | 0.88 |
| 100 Random Average clustering coefficient | 0.01±0.00 | 0.02±0.01 | 0.03±0.00 | 0.01±0.00 | 0.02±0.00 | 0.02±0.00 |
| 100 random Modularity | 0.81±0.01 | 0.56±0.01 | 0.34±0.00 | 0.65±0.01 | 0.40±0.00 | 0.59±0.00 |

was heavily polluted with Cd, Pb, and Zn [19]. Shen (2023) revealed notable enrichment of Cd and Hg in adjacent soils [20]. In this study, Cr content in MC was significantly higher than that in NC, suggesting that Cr from CG entered the surrounding soil through atmospheric deposition and wind transport, which was consistent with the results of Lu et al [45]. The contents of Cr, Cu, As, Pb and Zn in MLS were significantly exceeded those in MLZ and MLX, which reason was that long-term storage of CG due to weathering and rainwater leaching promoted the activity of harmful heavy metals in CG, some of them were dissolved into the water body or soil with precipitation and infiltration, continued to migrate and enrich, resulting in the increase of heavy metal content in the soil [13,14], and it showed a decreasing trend with the increase distance from CG dump [19,22], aligning with observations by Lu et al. [21]. We found that pH value in MC, MLS, MLZ, and MLX was significantly lower than that of NC and NL, which was due to the acidic wastewater containing heavy metal ions through dusting, rainfall, or infiltration produced by oxidation of sulfides in CG into the soil [46,47], which was similar to the results of Li et al [48]. Due to the increase in soluble salt ions and heavy metal ions in soil [15], the EC value had increased. Because the background quality and balance of heavy metals in the soil had changed and the original soil structure had been destroyed, soil nutrient content changed. For instance, we found the contents of TN, TP and AK in MC were significantly lower than those in NC, whereas SOC and TN in MLS were significantly higher than those in MLZ. This may be related to the content of carbon, nitrogen, phosphorus and potassium elements and their release rates from CG [49,50], also related to atmospheric deposition and leaching processes.

## 4.2 Influence of CG dump on soil bacterial community structure

Soil microbial community structure plays an important role in determining soil quality [51], with its community structure and distribution reflecting the quality of soil environment. Previous studies have demonstrated that the soil bacteria abundance around CG mountains is low [52]. In this study Proteobacteria was an important and prevalent bacteria in different disturbed areas, accounting for over 20% of relative abundance, which attributed to its metabolic versatility, environmental adaptability, competitive superiority in complex ecosystems [53], the finding consistent with the those reported by Hong et al [54]. Notably, the Proteobacteria in NL and LFA were significantly higher than those in MC and NC. We also found there were great differences in the top 20 bacterial genera with higher relative abundance in different regions. For example, the relative abundance of KD4–96 was the highest in MLS, is about 5%, while subgroup_6 was the highest in NC. The study revealed significantly reduced bacterial diversity and richness in MC compared to NC. with similar patterns observed between MLS and MLX in LFA. The reason should be that the difference of river material composition in different areas, the closer the distance to the CG dump, the greater the impact, and the mixed pollution of heavy metals soil will reduce

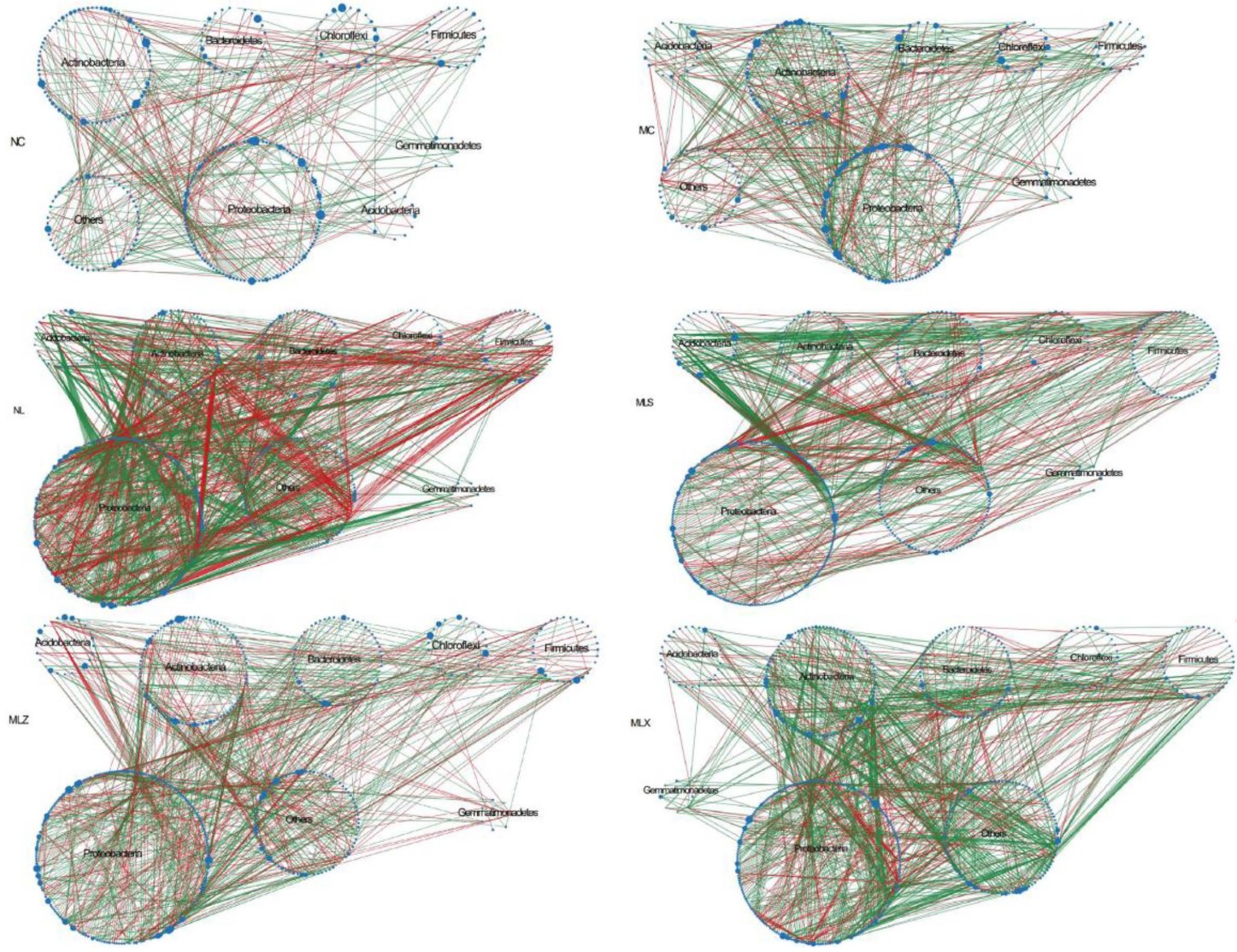

**Fig 8. Molecular ecological network of soil bacteria in different disturbed areas of CG dump.** Note: The size of the dots represents the degree of the nodes; Red represents positive correlation, green represents negative correlation, and the blue circle size represents the degree size value.

the bacterial diversity in soil [55]. These findings indicated that bacterial community structure in different disturbed areas of CG dump exhibit distinct responses to atmospheric dust fall and rainwater leaching.

### 4.3 Influence of CG dump on soil bacterial molecular ecological network

Microbial species play crucial roles in the natural environment by facilitating material cycling, energy flow, and environmental purification through complex network systems [55,56]. Within these networks, positive correlations between species indicated that the ecological niche of the species was the same or had a symbiotic relationship, while a negative correlation represented a competitive or predation relationship [57]. In this study six constructed networks were dominated by positive correlation, demonstrating that the bacteria in different disturbed areas were mainly cooperative while the competition relationship was weak. These results showed that the disturbance of CG dump led to the decrease of

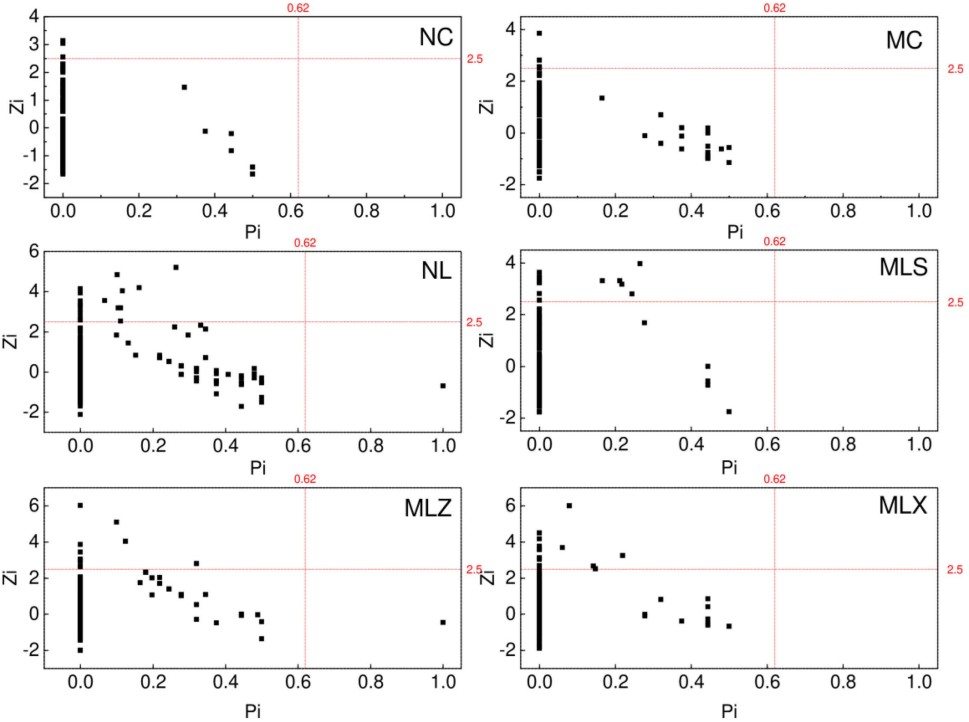

**Fig 9. Topological properties of soil bacteria in different disturbed areas of CG.** Note: Pi denotes inter-module connectivity and Zi denotes intra-module connectivity; the dot of Pi >0.62 and Zi > 2.5 represents network hub, Pi ≤ 0.62 and Zi > 2.5 represents module hub), the dot of Pi > 0.62 and Zi ≤ 2.5 represents connector node, and the dot of Pi ≤ 0.62 and Zi ≤ 2.5 represents peripheral node.

**Table 5. Module hub of soil bacterial network in different disturbed areas of CG dump.**

| | Number | Genus | Phyla |
|---|---|---|---|
| **NC** | 3 | Verrucomicrobia, S085, mle1–27 | Chloroflexi, Proteobacteria |
| **MC** | 5 | Subgroup_18, uncultured_f_Ilumatobacteraceae, MB-A2-108, Psychroglaciecola, Pseudomonas | Acidobacteria, Actinobacteria, Proteobacteria |
| **NL** | 17 | Elev-16S-573, Subgroup_10,0319-7L14, uncultured_f_Ilumatobacteraceae, unclassified_f_Micromonosporaceae, Flavobacteriaceae, RBG-13-54-9, TK10, BRH-c20a, Longimicrobiaceae, CPR2, Candidatus_Paracaedibacter, Devosia, Ketogulonicigenium, uncultured_f_Polyangiaceae, UASB-TL25, Dechlorosoma | Acidobacteria,Actinobacteria,Bacteroidetes, Chloroflexi, Firmicutes, Gemmatimonadetes, Proteobacteria, Patescibacteria |
| **MLS** | 12 | Leptothrix,uncultured_f_Desulfobulbaceae, Brevundimonas, Proteiniclasticum, UBA12409, ADurb.Bin180, unclassified_f_Parachlamydiaceae,Armatimonadales, Subgroup_23, uncultured_f_Holophagaceae, DS-100, Blastocatella | Proteobacteria, Firmicutes,Dependentiae Chloroflexi, Chlamydiae, Armatimonadetes, Acidobacteria |
| **MLZ** | 12 | Elev-16S-573, Subgroup_18, Rhizocola, unclassified_p_Actinobacteria,Gillisia, Leptolyngbya_LEGE-06070, uncultured_f_Peptostreptococcaceae,Rhodovastum, Roseomonas, uncultured_f_Reyranellaceae, Methyloceanibacter, OM27_clade | Acidobacteria, Actinobacteria, Bacteroidetes, Cyanobacteria, Firmicutes, Proteobacteria |
| **MLX** | 13 | uncultured_c_Acidimicrobiia, Acidothermus, Actinotale, Aeromicrobium, g_uncultured _c_Thermoleophilia, MWH-CFBk5, cvE6, OM190, g_uncultured_f_Azospirillaceae, Thioclava, Desulfuromonas, Polaromonas, Luteolibacter | Actinobacteria, Bacteroidetes, Chlamydiae, Planctomycetes, Proteobacteria, Verrucomicrobia |

the competition interactions among bacterial taxa in the surrounding soil and the enhancement of the synergistic effect. The distribution of bacteria in soil was also influenced by their ecological amplitude [58]. Species with broader ecological ranges demonstrate greater environmental tolerance and resource utilization capacity, enabling wider distribution [59]. Notably, the relative abundance of Actinomycetes and Chlamydiae in MLS was 6.42% and 0.19% respectively, while only Chlamydiae emerged as the key nodes in the bacterial network structure in MLS, which indicating that the key functional flora in the molecular ecological network of soil bacteria could not be identified solely based on relative abundance [60]. No common nodes were found in the bacterial network structure of different disturbed areas, which further indicated that there were great differences in different disturbed areas of CG dumps. In addition, the average network path was smallest in the MLS, which indicated that the soil bacteria were highly sensitive, environmental disturbances can rapidly propagate through the entire bacterial network, increasing structural instability [61].

## 4.4 Relationship between bacterial community and physical and chemical properties in the soil around CG dump

Microbial community structure is closely linked to soil characteristics [62]. For instance, variations in organic carbon content can drive geographical differences in soil microbial distribution [63], heavy metal concentrations [64,65], soil pH, and electrical conductivity (EC) [66,67] significantly influence the diversity and composition of soil bacterial communities. Studies have demonstrated that the long-term accumulation of CG alters soil physicochemicals properties, thereby affecting the uniqueness and diversity of bacterial community structure [2]. Zhu et al. (2023) observed that CG dump reduced soil pH and influenced the microbial community's capacity to degrade environmental pollutants [5]. Li et al. (2021) reported that bacterial abundance and diversity could enhance at copper concentrations ≤100 mg·kg$^{-1}$ but suppress at levels ≥500 mg·kg$^{-1}$ [68]. Additionally, Pan et al. (2020) identified soil available potassium and acid-extractable Pb as the primary factors shaping bacterial community structure [29]. In this study, redundancy analysis (RDA) revealed that heavy metals (As, Pb, Cu, and Cd), nutrients (SOC and TP), and EC values significantly influenced bacterial communities in different disturbed areas. These findings diverge from those of Hong et al. [54] but align with the results of Mei et al. [69], which may be related to the differences in CG composition and regional environmental conditions.

## 5 Conclusions and recommendations

According to our study, we were able to draw the following conclusions: (1) The response of soil bacteria to environmental disturbance and the stability of community structure were different in different disturbed areas. The cooperation of soil bacteria in different disturbed areas was dominant, and the scale of soil bacteria network in MLS was larger and the relationship between species was more complex. (2) The higher abundances of some bacterial communities play an important role in the molecular ecological network of soil bacteria in this study area, but the ecological function of lower abundances of bacteria in the bacterial network cannot be ignored. (3) The diversity of soil bacterial community was significantly different between the atmospheric dustfall area and the leachate flow area. The long-term storage of CG decreased the diversity of soil bacterial community of MC and MLS, SOC, TP, heavy metals As, Pb, Cu, Cd and EC in the soil in different disturbed areas had significant effects on soil microbial community structure.

However, there are still limitations: (1) The findings may be constrained by the geographic scope and temporal scale of sampling. The results may not fully capture long-term dynamics or broader ecosystem responses. (2) Although the study highlights the significant role of higher relative abundance bacterial communities in the molecular ecological network, the ecological functions of lower relative abundance bacteria remain underexplored. Their potential roles such as keystone species require further validation through metagenomics or other functional analyses. (3) Interactions between environmental factors were not quantified, and non-biotic influences (e.g., climate fluctuations or vegetation type) were not accounted for, potentially limiting the generalizability of the conclusions. In the future, our research should expand spatio-temporal sampling, integrate multi-omics approaches for deeper functional insights, and employ controlled experiments to isolate the effects of individual environmental drivers.

## Supporting information

**S1 Data.**
(XLS)

## Author contributions

**Conceptualization:** Bianhua Zhang, Dongsheng Jin, Qiang Zhang.

**Data curation:** Bianhua Zhang, Huijuan Bo, Wei Wang.

**Funding acquisition:** Dongsheng Jin, Qiang Zhang.

**Investigation:** Bianhua Zhang, Huijuan Bo, Wei Wang.

**Methodology:** Bianhua Zhang, Huijuan Bo, Wei Wang.

**Project administration:** Dongsheng Jin.

**Supervision:** Dongsheng Jin, Qiang Zhang.

**Writing – original draft:** Bianhua Zhang.

**Writing – review & editing:** Dongsheng Jin.

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
