## [Decision Letter · Decision Letter 0]

26 Mar 2025

Dear Dr. Jin,

In particular:

language errors should be removed (e.g. "areasf coal" => "areas of coal" (the title of Fig. 6)),presentation problems should be fixed (e.g. quality of Fig. 8, Fig. 9, precision of the title of section 1.4 and figures: 1,5,9; presentation of www links; format of references),experimental setup should be described in a more detailed way,statistical analysis needs to be presented with more details,analysis of measurement errors is missing,"Limitations of the study" section would increase the value of the text.

We look forward to receiving your revised manuscript.

Kind regards,

Maciej Huk, Ph.D.

Academic Editor

PLOS ONE

“National Key R&D Program�2020YFC1806501-2Supported by fundamental Research Program of Shanxi Province,202203021221223; Major Project of Science and Technology in Shanxi Province�202201140401028.”

6. "Please include your tables as part of your main manuscript and remove the individual files. Please note that supplementary tables (should remain/ be uploaded) as separate "supporting information" files

Reviewers' comments:

Reviewer's Responses to Questions

**Comments to the Author**

1. Is the manuscript technically sound, and do the data support the conclusions?

Reviewer #1: Yes

Reviewer #2: Yes

Reviewer #3: Yes

2. Has the statistical analysis been performed appropriately and rigorously?

Reviewer #1: Yes

Reviewer #2: Yes

Reviewer #3: Yes

3. Have the authors made all data underlying the findings in their manuscript fully available?

Reviewer #1: Yes

Reviewer #2: Yes

Reviewer #3: Yes

4. Is the manuscript presented in an intelligible fashion and written in standard English?

Reviewer #1: Yes

Reviewer #2: Yes

Reviewer #3: No

Reviewer #1: Line 168: All statistical tests used in the analyses should be detailed in the Statistical Analyses section of the Methods, in addition to naming the software. Additionally, the function and purpose of each statistical test should be clearly described.

Classification for Nodes role e.g. peripheral nodes, module hubs, connectors and network hubs based on Zi & Pi score are to be presented in the statistical analyses section of the Methods.

Line 168: Data sorting and analysis. There was no description of data sorting. The title is to be revised.

Line 169: The proper citation for the statistical software, including the publisher's name and the specific name of the statistical test for the variance analysis, is to be stated.

Line 171: One or two-tailed test is to be stated.

Line 173: The URL link is spaced out. To revise to http://ieg4.rccc.ou.edu/mena/

Line 175-177: The sentence requires revision. The version of the software is to be stated. Network Analyzer is a built-in plugin/tool within Cytoscape. The links for all the software is to be provided. Origin is useful for statistical analysis and high-quality visualization of bacterial network data; however, it is not designed for network construction or topological parameter computation.

Line 207-208 the figures are to be spaced from the phylum name.

Subtitle 2.2 – 2.5: to be spaced out from the previous paragraph.

Line 227: For Shannon Index: Species diversity and evenness.

Line 228: chao1 index (cap C)

Line 246: Principal Coordinates Analysis (PCoA) is to be stated.

Line 249, Line 272, Line 277, Line 280, Line 289, Line 290, Line 296, Line 298, Line 300: There were some typographical errors e,.g NL,MLS MLZ,MLX , ,while, Pi>0.62),which function.Generally Table 6,these Chlamydiae,Dependentiae and6.42% (to space out)

Line 261: MC, (comma to be replaced with dot or word while).

Line 268: Two decimal point for R^2=0.8

Line 425: MLS. SOC (comma)

Line 302: Discussion

Figure 4: *, ** and *** to be denoted/described in the figure footnote.

Figure 8: Too small, blur and hard to visualize.

Figure 9: The diagram is a bit small. Axis are to be denoted in the footnote.

Network topological properties (metrics table), taxa, modules, validation could be presented apart from Figure 9.

References did not conform to the journal format.

Reviewer #2: 1- The abstract needs to be rewritten to clarify the results of the work with numerical values that confirm its efficiency.

2- The introduction is not numbered, and the researcher's contributions are not clearly stated. The introduction also requires a paragraph explaining the structure of the research in all its sections.

3- A section detailing a related works of the most important research presented in the same field for a period of time extending between 2020-2025.

4- Discussing the results clearly reflects the researcher's understanding of the work he has presented.

Reviewer #3: ### **Review Comments to the Author**

Dear Authors,

Thank you for the opportunity to review your manuscript titled *“Long-term coal gangue dump regulates bacterial communities in different disturbance areas: Evidence mostly from diversity and network complexity.”* The study addresses an important environmental issue and provides valuable insights into the effects of long-term coal gangue accumulation on soil bacterial communities using molecular ecological network analysis and high-throughput sequencing. Overall, your manuscript presents meaningful data and offers several noteworthy findings. Below are my detailed comments and suggestions for improvement:

---

### **Strengths:**

1. **Novelty & Relevance**: The focus on bacterial community dynamics across different disturbance zones of a coal gangue dump is timely and relevant, especially with growing interest in soil microbial ecology and environmental pollution.

2. **Methodology**: The experimental design is sound, with appropriate sample collection zones (NC, MC, NL, MLS, MLZ, MLX) that capture varying degrees of disturbance. The use of 16S rRNA sequencing, diversity indices, and ecological network analysis adds depth to the study.

3. **Data Interpretation**: The correlation between microbial patterns and soil physicochemical properties is well-supported, especially the finding that SOC, As, and Pb significantly influence community structure.

---

### **Areas for Improvement:**

#### **1. English Language and Clarity:**

- While the manuscript is generally understandable, it contains multiple grammatical issues, awkward phrasing, and non-standard expressions.

- For example:

- “we took the coal gangue storage site… as the research object” → “we selected the coal gangue storage site… as the study area.”

- “bacteria and bacteria” → “interactions among bacterial taxa” or “microbial taxa.”

- A professional English language editing service is highly recommended to enhance readability and clarity.

#### **2. Abstract:**

- The abstract is informative but overly detailed. Consider condensing it by summarizing key results more concisely.

- Please clarify statements such as “flora under the phyla with lower abundance may be the key node” for better reader comprehension.

#### **3. Figures and Tables:**

- Figures are informative, but some require better labeling and legends for standalone interpretation.

- Ensure all abbreviations (e.g., MC, NC, SOC) are clearly defined when first used in each figure/table and in the main text.

#### **4. Discussion:**

- The discussion section is descriptive but would benefit from a clearer separation between discussion and results.

- Consider integrating more critical comparisons with related literature to emphasize how your findings advance existing knowledge.

#### **5. Conclusion:**

- The conclusion summarizes the findings well, but a final remark on the potential implications for ecological remediation or future research directions would enhance its impact.

---

### **Minor Suggestions:**

- Double-check the formatting of references to ensure consistency.

- Clarify whether ethics approval or permits were required, especially for field research.

- Include DOIs for all references where available.

---

### **Overall Recommendation:**

Your study contributes meaningfully to the understanding of microbial responses to coal gangue disturbances. After **addressing language issues and improving clarity in the discussion**, the manuscript will be well-positioned for publication.

**Do you want your identity to be public for this peer review?** For information about this choice, including consent withdrawal, please see our Privacy Policy

Reviewer #1: No

Reviewer #2: No

Reviewer #3: No

---

## [Author Response · Author response to Decision Letter 1]

13 May 2025

Dear Editor,

We are grateful to you and the other reviewers for your critical comments and thoughtful suggestions on our manuscript. We have carefully revised the manuscript based on your feedbacks. The revised portions of the manuscript are shown in red. We hope the revised manuscript satisfies the standards of the journal.

Below, we provide point-by-point responses to reviewer#1, reviewer#2 and reviewer#3’s comments and questions. We hope that these revisions are satisfactory and the revised manuscript could meet the publishing standards of the PLoS One.

Thank you very much for your assistance with our paper.

1.Language errors should be removed (e.g. "areasf coal" => "areas of coal" (the title of Fig. 6)),

Response: Thanks for your suggestion very much, all typographical and grammatical errors have been corrected (e.g., "areas of coal" in Figure 6’s title has been revised to “areas of coal”). The text has been carefully proofread to ensure clarity and accuracy.

2.Presentation problems should be fixed (e.g. quality of Fig. 8, Fig. 9, precision of the title of section 1.4 and figures: 1,5,9; presentation of www links; format of references),

Response: Thanks for your suggestion very much, the quality of Fig 8 and Fig 9 has been enhanced for better readability. The title of section 1.4 and the captions of Figures 1, 5, and 9 have been refined for precision. Web links (URLs) have been formatted consistently, and reference formatting has been standardized.

3.Experimental setup should be described in a more detailed way.

Response: Thanks for your suggestion very much, a more detailed description of the experimental setup has been described, including methods, equipments, and related references etc. in section of 2.

4.Statistical analysis needs to be presented with more details

Response: Thanks for your suggestion very much, additional details on statistical methods, including significance tests and data processing have been included in the section of 2.4(data statistical analysis).

5.Analysis of measurement errors is missing.

Response: Thanks for your suggestion very much, we have added note in table 1 and 2(The data in the table are mean±standard error, Different lowercase letters within a column indicate significant differences at the 5% level. Significant difference was performed between that in different disturbed areas of soils.) and in the section of 2.4.

6."Limitations of the study" section would increase the value of the text.

Response: Thanks for your suggestion very much, a new section has been added to outline the study’s limitations in section of 5 in line 394-406, providing context for the findings and suggesting directions for future research.

Responses to Reviewer #1:

Dear Editor,

We are grateful to you and the other reviewers for your critical comments and thoughtful suggestions on our manuscript. We have carefully revised the manuscript based on your feedbacks. The revised portions of the manuscript are shown in red. We hope the revised manuscript satisfies the standards of the journal.

Below, we provide point-by-point responses to your comments and questions. We hope that these revisions are satisfactory and the revised manuscript could meet the publishing standards of the PLoS One.

Thank you very much for your assistance with our paper.

Line 168: All statistical tests used in the analyses should be detailed in the Statistical Analyses section of the Methods, in addition to naming the software. Additionally, the function and purpose of each statistical test should be clearly described. Classification for Nodes role e.g. peripheral nodes, module hubs, connectors and network hubs based on Zi & Pi score are to be presented in the statistical analyses section of the Methods.

Response: Thanks for your suggestion very much, we added a detailed description of all statistical tests used, including their functions and purposes. Clarified the classification of node roles (peripheral nodes, module hubs, connectors, network hubs) based on Zi and Pi scores, explicitly stated in the section of 2.4.

Line 168: Data sorting and analysis. There was no description of data sorting. The title is to be revised.

Response: Thanks for your suggestion very much, we revised the section title to better reflect its content (now titled “Data statistical analysis”).

Line 169: The proper citation for the statistical software, including the publisher's name and the specific name of the statistical test for the variance analysis, is to be stated.

Response: Thanks for your suggestion very much, we provided full citations for statistical software, including publisher names (e.g., “Version 26.0, SPSS Inc., Chicago, IL, USA”) in section of 2.4.

Line 171: One or two-tailed test is to be stated.

Response: Thanks for your suggestion very much, we revised it in section of 2.4. to “means were declared significant at (p<0.05) according to Tukey’s Honesty Test” in line 171-172.

Line 173: The URL link is spaced out. To revise to http://ieg4.rccc.ou.edu/mena/

Response: Thanks for your suggestion very much, The URL link has revised to to http://ieg4.rccc.ou.edu/mena/ in line 181.

Line 175-177: The sentence requires revision. The version of the software is to be stated. Network Analyzer is a built-in plugin/tool within Cytoscape. The links for all the software is to be provided. Origin is useful for statistical analysis and high-quality visualization of bacterial network data; however, it is not designed for network construction or topological parameter computation.

Response: Thanks for your suggestion very much, we revised the sentence to accurately describe tools: Stated the version of Cytoscape and clarified that Network Analyzer is a built-in plugin. Removed the incorrect implication about Origin’s role in network construction, added its message Origin (Version9.0, OriginLab Corporation, Canada) in section of 2.4.

Line 207-208 the figures are to be spaced from the phylum name. Subtitle 2.2 – 2.5: to be spaced out from the previous paragraph.

Response: Thanks for your suggestion very much, Ensured proper spacing between phylum names and figures. Adjusted spacing between subtitles (2.2-2.5) and preceding paragraphs.

Line 227: For Shannon Index: Species diversity and evenness.

Response: Thanks for your suggestion very much, we revised the Shannon Index is to characterzize species diversity and evenness in line 248.

Line 228: chao1 index (cap C)

Response: Thanks for your suggestion very much, Corrected “chao1” to “Chao1” in line 234.

Line 246: Principal Coordinates Analysis (PCoA) is to be stated.

Response: Thanks for your suggestion very much, “Principal Coordinates Analysis (PCoA)” has stated in section of 2.4 in line 172-174.

Line 249, Line 272, Line 277, Line 280, Line 289, Line 290, Line 296, Line 298, Line 300: There were some typographical errors e,.g NL,MLS MLZ,MLX , ,while, Pi>0.62),which function.Generally Table 6,these Chlamydiae,Dependentiae and6.42% (to space out)

Response: Thanks for your suggestion very much, all typographical errors and formatting have corrected.

Line 261: MC, (comma to be replaced with dot or word while).

Response: Thanks for your suggestion very much, we replaced comma with “while” in Line 261.

Line 268: Two decimal point for R^2=0.8

Response: Thanks for your suggestion very much, we added a second decimal to R² values (now “R² = 0.80”) in line 264.

Line 425: MLS. SOC (comma)

Response: Thanks for your suggestion very much, we added a comma after “MLS” (now “MLS, SOC”).

Line 302: Discuss

Response: Thanks for your suggestion very much, we revised” Discuss” to” Discussion” in line 283.

Figure 4: *, ** and *** to be denoted/described in the figure footnote.

Response: Thanks for your suggestion very much, added a legend to denote statistical significance markers (Note: ***represents p < 0.001, ** represents p < 0.01, * represents p < 0.05) of Figure 4.

Figure 8: Too small, blur and hard to visualize.

Response: Thanks for your suggestion very much, we have enlarged the figure and enhanced its resolution to ensure better visibility and eliminate blurriness, and provided their PDF.

Figure 9: The diagram is a bit small. Axis are to be denoted in the footnote. Network topological properties (metrics table), taxa, modules, validation could be presented apart from Figure 9.

Response: Thanks for your suggestion very much, the diagram has been resized for improved readability. Axis notations have been denoted in the footnote. Network topological properties taxa were in table 5.

References did not conform to the journal format.

Response: Thanks for your suggestion very much, we have revised the paper’s format.

Reviewer #2:

Dear Editor,

We are grateful to you and the other reviewers for your critical comments and thoughtful suggestions on our manuscript. We have carefully revised the manuscript based on your feedbacks. The revised portions of the manuscript are shown in red. We hope the revised manuscript satisfies the standards of the journal.

Below, we provide point-by-point responses to your comments and questions. We hope that these revisions are satisfactory and the revised manuscript could meet the publishing standards of the PLoS One.

Thank you very much for your assistance with our paper.

1- The abstract needs to be rewritten to clarify the results of the work with numerical values that confirm its efficiency.

Response: Thanks for your suggestion very much, we have rewritten the abstract to clarify the key results, including specific numerical values that highlight significance of our findings in line 28-57.

2- The introduction is not numbered, and the researcher's contributions are not clearly stated. The introduction also requires a paragraph explaining the structure of the research in all its sections.

Response: Thanks for your suggestion very much, the introduction has been restructured with numbered for better clarity in line 60. The researchers’ contributions are now explicitly stated. A new paragraph has been added to outline the structure of the paper in line 107-120.

3- A section detailing a related works of the most important research presented in the same field for a period of time extending between 2020-2025.

Response: Thanks for your suggestion very much, we have been added the most relevant and recent literature in the coal gangue field, with a focus on advancements between 2020 and 2025 to the introduction in line 65-89.

4- Discussing the results clearly reflects the researcher's understanding of the work he has presented.

Response: Thanks for your suggestion very much, the discussion has been refined to more clearly articulate the implications of our findings, ensuring a deeper interpretation of the results.

Reviewer #3:

Dear Editor,

We are grateful to you and the other reviewers for your critical comments and thoughtful suggestions on our manuscript. We have carefully revised the manuscript based on your feedbacks. The revised portions of the manuscript are shown in red. We hope the revised manuscript satisfies the standards of the journal.

Below, we provide point-by-point responses to your comments and questions. We hope that these revisions are satisfactory and the revised manuscript could meet the publishing standards of the PLoS One.

Thank you very much for your assistance with our paper.

1. English Language and Clarity

- While the manuscript is generally understandable, it contains multiple grammatical issues, awkward phrasing, and non-standard expressions.

- For example:

- “we took the coal gangue storage site… as the research object” → “we selected the coal gangue storage site… as the study area.”

- “bacteria and bacteria” → “interactions among bacterial taxa” or “microbial taxa.”

- A professional English language editing service is highly recommended to enhance readability and clarity.

Response: Thanks for your suggestion very much, the manuscript has undergone thorough language editing to correct grammatical errors and enhance overall readability by inviting a professional editing service to ensure linguistic precision. Some expression has been revised based on your suggestion. Such as “We took the coal gangue storage site… as the research object” has revised to “We selected the coal gangue storage site… as the study area.” “Bacteria and bacteria” revised to “Interactions among bacterial taxa.”

2. Abstract

- The abstract is informative but overly detailed. Consider condensing it by summarizing key results more concisely.

- Please clarify statements such as “flora under the phyla with lower abundance may be the key node” for better reader comprehension.

Response: Thanks for your suggestion very much, the abstract has been revised for conciseness while retaining key results. Ambiguous statements have been modified its expression for better comprehension. (e.g., “flora under the phyla with lower abundance may be the key node” has been revised to “the bacteria with higher relative abundance may be not the key node in the bacterial molecular ecological network, while bacteria with the lower relative abundance maybe” in line 49-51).

3. Figures and Tables:

- Figures are informative, but some require better labeling and legends for standalone interpretation.

- Ensure all abbreviations (e.g., MC, NC, SOC) are clearly defined when first used in each figure/table and in the main text.

Response: Thanks for your suggestion very much, we enlarged for better visibility for clarity and added footnotes to Figures. All abbreviations (e.g., MC, NC, SOC) are now clearly defined upon first use in both figures and text.

4. Discussion:

- The discussion section is descriptive but would benefit from a clearer separation between discussion and results.

- Consider integrating more critical comparisons with related literature to emphasize how your findings advance existing knowledge.

Response: Thanks for your suggestion very much, the section has been reorganized to clearly distinguish between results and discussion, additional comparisons with recent literature have been integrated to emphasize how our findings advance current knowledge in section of 4.1-4.4.

5. Conclusion:

- The conclusion summarizes the findings well, but a final remark on the potential implications for ecological remediation or future research directions would enhance its impact.

Response: Thanks for your suggestion very much, the conclusion now includes our study limitation and future research directions in line 404-416.

Minor Suggestions:

- Double-check the formatting of references to ensure consistency.

Response: Thanks for your suggestion very much, we have carefully revised the formatting of references to ensure consistency.

- Clarify whether ethics approval or permits were required, especially for field research.

Response: Thanks for your suggestion very much, ethical approval or licensing is not required for the study.

- Include DOIs for all references where available.

Response: Thanks for your suggestion very much, we have included DOIs for all references where available except for references of 9, 30, and 54 because they have no DOI.

We hope these revisions have addressed all concerns raised by the reviewers and significantly strengthened the manuscript. Thank you for your valuable time and insightful suggestions.

Sincerely

Dongsheng Jin

Shanxi Agriculture University

---

## [Decision Letter · Decision Letter 1]

1 Jun 2025

Dear Dr. Jin,

Thank you for submitting your manuscript to PLOS ONE. The manuscript was analysed by four Reviewers including me as an Academic Editor (Reviewer #4). After careful consideration, we feel that it has merit but does not fully meet PLOS ONE’s publication criteria as it currently stands. Therefore, we invite you to submit a revised version of the manuscript that addresses the points raised during the review process.

In particular: minor language problems need to be removed before publication.

We look forward to receiving your revised manuscript.

Kind regards,

Maciej Huk, Ph.D.

Academic Editor

PLOS ONE

Journal Requirements:

Reviewers' comments:

Reviewer's Responses to Questions

**Comments to the Author**

Reviewer #1: All comments have been addressed

Reviewer #2: (No Response)

Reviewer #3: All comments have been addressed

Reviewer #4: All comments have been addressed

2. Is the manuscript technically sound, and do the data support the conclusions?

Reviewer #1: Yes

Reviewer #2: (No Response)

Reviewer #3: Yes

Reviewer #4: Yes

3. Has the statistical analysis been performed appropriately and rigorously?

Reviewer #1: Yes

Reviewer #2: (No Response)

Reviewer #3: Yes

Reviewer #4: I Don't Know

4. Have the authors made all data underlying the findings in their manuscript fully available?

Reviewer #1: Yes

Reviewer #2: (No Response)

Reviewer #3: Yes

Reviewer #4: Yes

5. Is the manuscript presented in an intelligible fashion and written in standard English?

Reviewer #1: Yes

Reviewer #2: (No Response)

Reviewer #3: Yes

Reviewer #4: No

Reviewer #1: Spacing/caps required for the following:

Line 139: area(LFA)

Line 187: Principal Coordinate Analysis

Line 189: Canoco 5.0, Canoco 5 Support Site www.canoco5.com

Line 197: Origin(Version9.0,OriginLab Corporation

Line 314: soil[13,14]

Line 315: distance[19,22]

Line 312: reference bracket ( ) to be replaced with [ ]

Fig to be replaced with Fig. or Figure throughout the manuscript.

Reviewer #2: (No Response)

Reviewer #3: **Review Comments to the Author:**

The revised manuscript titled *“Long-term coal gangue dump regulates bacterial communities in different disturbance areas: Evidence mostly from diversity and network complexity”* presents a well-executed study that investigates the effects of coal gangue accumulation on soil bacterial communities across varying disturbance zones. I commend the authors for their comprehensive response to the previous round of reviewer comments and for significantly improving the manuscript's clarity, structure, and analytical depth.

### Strengths of the Manuscript:

1. **Technical Soundness**: The experimental design is robust, incorporating clearly defined zones with appropriate sampling replication. The use of 16S rDNA sequencing, physicochemical analysis, and network ecology tools is methodologically sound and well-justified.

2. **Data-Driven Conclusions**: The conclusions are appropriately supported by the data. The authors provide clear statistical evidence linking soil bacterial diversity and community structure to environmental variables, including heavy metals and soil nutrients.

3. **Statistical Rigor**: The statistical analyses (ANOVA, RDA, PCoA, and ecological network metrics) are appropriate and described with sufficient detail. The use of established software tools like SPSS, Canoco, and Cytoscape adds confidence to the results.

4. **Data Availability**: All data supporting the findings are fully available within the manuscript and supplementary materials, meeting PLOS ONE’s open data policy.

5. **Presentation and Language**: The manuscript is clearly structured and generally well-written. The authors have incorporated professional English editing, and the readability has improved significantly from the original version. While a few minor grammatical or stylistic issues remain, they do not affect overall comprehension.

### Suggestions for Further Improvement:

* **Language Polishing**: A final proofreading pass could further enhance clarity, especially in the abstract and discussion sections where some phrasing remains slightly awkward (e.g., “may be not the key node” → “may not be the key node”).

* **Figure and Table Legends**: Ensure that all abbreviations and statistical markers (e.g., \*, \*\*, \*\*\*) are clearly defined in figure legends for standalone comprehension.

* **Literature Context**: The introduction now includes recent studies (2020–2025), which is excellent. A few more comparative insights in the discussion, especially contrasting findings with studies from similar post-mining contexts globally, could strengthen the interpretation further.

### Final Assessment:

The authors have successfully addressed all major concerns raised during the previous review. The manuscript now meets the standards of scientific rigor, transparency, and clarity expected by *PLOS ONE*. I support the publication of this work.

Reviewer #4: >>> 1. Language problems:

1.1 there was no significant differences => there were no significant differences

1.2 while bacteria with the lower relative abundance maybe.

=> while bacteria with the lower relative abundance might have been.

1.3 "From Table 5 and Fig 8, the positive correlation between bacteria in different disturbed areas accounted for more than 58%, indicating that cooperative cooperation was dominant and competition was weak."

=>

"As shown in Table 5 and Figure 8, positive correlations between bacteria in different disturbed areas accounted for more than 58%, indicating that cooperative interactions were dominant, while competitive interactions were weak."

1.4 "Species play an important role in the process of material circulation, energy flow, environmental pollution indication and purification by forming complex network systems In the natural environment"

In => in

>>> 2. Presentation problems: not detected

>>> 3. Other problems: not detected

>>> Recommendation: Minor rework

===EOT===

**Do you want your identity to be public for this peer review?** For information about this choice, including consent withdrawal, please see our Privacy Policy

Reviewer #1: No

Reviewer #2: No

Reviewer #3: No

Reviewer #4: No

---

## [Author Response · Author response to Decision Letter 2]

7 Aug 2025

Dear Editor,

We are grateful to you and the other reviewers for your critical comments and thoughtful suggestions on our manuscript. We have carefully revised the manuscript based on your feedbacks. The revised portions of the manuscript are shown in red. We include the figure legends and tables within the manuscript. We hope the revised manuscript satisfies the standards of the journal.

Below, we provide point-by-point responses to reviewer#1, reviewer#3 and reviewer#4’s comments and questions. We hope that these revisions are satisfactory and the revised manuscript could meet the publishing standards of the PLoS One.

Thank you very much for your assistance with our paper.

Responses to Reviewer #1:

Spacing/caps required for the following:

Line 139: area(LFA)

Response: Thanks for your suggestion very much, we have added spacing and corrected it to area (LFA) in Line 145.

Line 187: Principal Coordinate Analysis

Response: Thanks for your suggestion very much, “Principal coordinate analysis” have corrected to “Principal Coordinate Analysis” in line 200.

Line 189: Canoco 5.0, Canoco 5 Support Site www.canoco5.com

Response: Thanks for your suggestion very much, we have added spacing and punctuation www.canoco5.com in line 203.

Line 197: Origin (Version9.0, Origin Lab Corporation

Response: Thanks for your suggestion very much, we have added spacing in line 210.

Line 314: soil[13,14]

Response: Thanks for your suggestion very much, we have added spacing and corrected to soil [13,14] in line 381.

Line 315: distance[19,22]

Response: Thanks for your suggestion very much, we have added spacing and corrected it in line 382.

Line 312: reference bracket ( ) to be replaced with [ ]

Response: Thanks for your suggestion very much, all reference brackets in the manuscript have replaced as required.

Fig to be replaced with Fig. or Figure throughout the manuscript.

Response: Thanks for your suggestion very much, “Fig” has been replaced with “Figure” consistently throughout the manuscript.

Responses to Reviewer #3:

Language Polishing: A final proofreading pass could further enhance clarity, especially in the abstract and discussion sections where some phrasing remains slightly awkward (e.g., “may be not the key node” → “may not be the key node”).

Response: Thanks for your suggestion very much, we have corrected awkward phrasing (e.g., “may be not the key node” → “may not be the key node” in line 18 and line 52) and refined sentence structures for improved readability, conducted a full proofreading pass to ensure consistency, grammatical accuracy, and fluency across the entire manuscript, especially in key sections such as the abstract and discussion.

Figure and Table Legends: Ensure that all abbreviations and statistical markers (e.g., \*, \*\*, \*\*\*) are clearly defined in figure legends for standalone comprehension.

Response: Thanks for your suggestion very much, all abbreviations used in figures and tables are now clearly defined in their respective legends to ensure standalone comprehension; symbols for statistical significance are now explicitly explained in each legend (e.g., *represents p < 0.05, ** represents p < 0.01, *** represents p < 0.001”).

Literature Context: The introduction now includes recent studies (2020–2025), which is excellent. A few more comparative insights in the discussion, especially contrasting findings with studies from similar post-mining contexts globally, could strengthen the interpretation further.

Response: Thanks for your suggestion very much, according to your suggestion, we have added comparative insights in the discussion (e.g., in line 371-373, 448-457).

Responses to Reviewer #4:

1. Language problems:

1.1 there was no significant differences => there were no significant differences

Response: Thanks for your suggestion very much, “there was no significant differences” has been corrected to "there were no significant differences" in line 236.

1.2 while bacteria with the lower relative abundance maybe.

=> while bacteria with the lower relative abundance might have been.

Response: Thanks for your suggestion very much, “while bacteria with the lower relative abundance maybe” has been corrected to “while bacteria with the lower relative abundance might have been” in line 19 and line 54.

1.3 "From Table 5 and Fig 8, the positive correlation between bacteria in different disturbed areas accounted for more than 58%, indicating that cooperative cooperation was dominant and competition was weak."

=>

"As shown in Table 5 and Figure 8, positive correlations between bacteria in different disturbed areas accounted for more than 58%, indicating that cooperative interactions were dominant, while competitive interactions were weak."

Response: Thanks for your suggestion very much, we checked the table, “From Table 5 and Fig 8” has been corrected to “As shown in Table 4 and Figure 8”in line 331.

1.4 "Species play an important role in the process of material circulation, energy flow, environmental pollution indication and purification by forming complex network systems In the natural environment"

In => in

Response: Thanks for your suggestion very much, “In the natural environment" has been corrected to “in the natural environment” in line 416.

We hope these revisions have addressed all concerns raised by the reviewers and significantly strengthened the manuscript. Thank you for your valuable time and insightful suggestions.

Sincerely

Dongsheng Jin

Shanxi Agriculture University

---

## [Decision Letter · Decision Letter 2]

27 Aug 2025

Long-term coal gangue dump regulates bacterial communities in different disturbance areas: Evidence mostly from diversity and network complexity

PONE-D-25-12223R2

Dear Dr. Jin,

We’re pleased to inform you that your manuscript has been judged scientifically suitable for publication and will be formally accepted for publication once it meets all outstanding technical requirements.

Kind regards,

Maciej Huk, Ph.D.

Academic Editor

PLOS ONE

Additional Editor Comments (optional):

Reviewers' comments:

Reviewer's Responses to Questions

**Comments to the Author**

Reviewer #4: All comments have been addressed

2. Is the manuscript technically sound, and do the data support the conclusions?

Reviewer #4: Yes

3. Has the statistical analysis been performed appropriately and rigorously?

Reviewer #4: Yes

4. Have the authors made all data underlying the findings in their manuscript fully available?

Reviewer #4: Yes

5. Is the manuscript presented in an intelligible fashion and written in standard English?

Reviewer #4: Yes

Reviewer #4: >>> 1. Language problems:

1.1 "relative abundance [60].No common nodes were found" => "relative abundance [60]. No common nodes were found" (missing space after the period)

>>> 2. Presentation problems:

2.1 Fig. 7, Table 3: RDA - each abbreviation should be defined befor its first use. In the manuscript RDA is defined much later than Fig 7 and Table 3 are appearing.

Also please consider :

Fig. 7: title: Relationship between bacteria and physical and chemical properties in the soil by redundancy analysis

Table 3: title: Results of RDA => Results of redundancy analysis

>>> 3. Other problems:

3.1 Table 1, Table 2: Note: "Significant difference was performed between that in different disturbed areas of soils."

was performed between that => was observed between that (?)

The meaning of this sentence is not clear. It is unclear to what element "between that" referes to.

>>> Recommendation: Accept (after removing the three minor problems listed above).

===EOT===

**Do you want your identity to be public for this peer review?** For information about this choice, including consent withdrawal, please see our Privacy Policy

Reviewer #4: No

---

## [Editor Report · Acceptance letter]

PONE-D-25-12223R2

PLOS ONE

Dear Dr. Jin,

I'm pleased to inform you that your manuscript has been deemed suitable for publication in PLOS ONE. Congratulations! Your manuscript is now being handed over to our production team.

Kind regards,

on behalf of

Dr. Maciej Huk

Academic Editor

PLOS ONE